# Effects of Model Grid Spacing for Warm Conveyor Belt (WCB) Moisture Transport into the Upper Troposphere and Lower Stratosphere (UTLS) — Part II: Eulerian Perspective

rait II. Eulerian Ferspective

Cornelis Schwenk <sup>1</sup> and Annette Miltenberger <sup>1</sup>

<sup>1</sup>Institute for Atmospheric Physics, Johannes Gutenberg University Mainz

**Correspondence:** Cornelis Schwenk (c.schwenk@uni-mainz.de)

Abstract. Warm conveyor belts (WCBs) are important features of extratropical cyclones that transport water vapor and hydrometeors into the upper troposphere and lower stratosphere (UTLS), influencing Earth's radiative budget. Previous studies have demonstrated that the horizontal grid spacing of numerical weather prediction (NWP) models influences modeled WCB properties such as ascent rates and diabatic heating. This two-part study investigates how model grid spacing affects the transport of moisture. We analyze two ICON model simulations of one North Atlantic WCB case study: a convection-parameterizing run at  $\sim$ 13 km and a convection-permitting run at  $\sim$ 3.5 km approximate grid spacing. Here, present the Eulerian perspective to complement the Lagrangian perspective from the first part of this study. We determine that the convection-parameterizing simulation produces a more humid UTLS in the WCB outflow. This results from (i) larger and fewer ice crystals, slowing the depletion of supersaturation, and (ii) the convection parameterization scheme, which injects excess vapor into the UTLS, when compared to the convection permitting simulation. The convection-permitting simulation experiences larger vertical velocities, which allows for the formation of thicker clouds with more graupel. Cloud-top temperatures are similar, yet the convection permitting simulation produces more outgoing long-wave radiation, which we can attribute to differences in UTLS vapor. Our findings indicate that convection-parameterizing simulations likely misrepresent moisture and hydrometeor transport by WCBs. This has implications for how global climate models simulate the radiative impact of WCBs and their potential influence on upper-level flow.

#### 1 Introduction

In the upper troposphere and lower stratosphere (UTLS), variations in water vapor concentration drive one of the most important positive climate feedbacks (Li et al., 2024; Held and Soden, 2000; Dessler et al., 2013, e.g.,), making water vapor a key greenhouse gas in this region (Schneider et al., 2010). Despite its critical role, the UTLS remains one of the least well-characterized parts of the atmosphere (Jeffery et al., 2022), and the processes governing its moisture content — including the

mechanisms of moisture transport — are still subject to considerable uncertainty (Petzold et al., 2020; Jeffery et al., 2022; Zahn et al., 2014; Guo and Miltenberger, 2025). The most prominent source of global UTLS moisture is deep convection in the tropics (Lee et al., 2019; Ueyama et al., 2023; Dauhut et al., 2018; Hassim and Lane, 2010; Corti et al., 2008; Gordon et al., 2024). However, warm conveyor belts (WCBs) have also been identified as a major source of moisture in the extratropical UTLS by experimental (Zahn et al., 2014) and climatological studies ((Guo and Miltenberger, 2025)).

WCBs are an intrinsic part of extratropical cylones (ETCs), where large-scale ascending air streams produce the typical elongated and comma-shaped cloud-bands associated with them. WCBs are important for mid-latitude weather patterns and the large amounts of precipitation formed by WCBs can be hazardous (Pfahl et al., 2014; Eckhardt et al., 2004). During the ascent of planetary boundary layer (PBL) air to the UTLS, WCBs produce mixed-phase clouds and precipitation and ultimately inject water vapor and hydrometeors into the UTLS (Madonna et al., 2014; Schwenk and Miltenberger, 2024). Their clouds also interact with radiation, making WCBs important when modeling Earth's radiative budget (Joos, 2019). For these reasons, it is important that numerical weather prediction (NWP) and global climate models (GCMs) accurately represent WCBs.

However, the correct representation of WCBs and in particular the moisture transport is difficult because WCB airstreams are a highly dynamic, multi-phase phenomenon that undergoes complex microphysical processes at each stage of their life (Binder et al., 2020; Forbes and Clark, 2003; Gehring et al., 2020; Hieronymus et al., 2025). Sensitivity experiments have shown that the choice of microphysical parameterization schemes significantly affects WCB ascent characteristics (Mazoyer et al., 2021, 2023), while the specific microphysical parameters within these schemes strongly influence diabatic processes during ascent (Hieronymus et al., 2022; Neuhauser et al., 2023; Forbes and Clark, 2003). Oertel et al. (2025); Schwenk et al. (2025) further demonstrated that modifying microphysical parameters in NWP models — such as the capacitance of ice and snow — even within plausible ranges, can substantially alter diabatic heating and ice content in the WCB outflow.

Collectively, these studies highlight the complexity of microphysical processes in WCBs and the major challenges they pose for accurate modeling. This is an important issue because the incorrect representation of WCBs has been linked to degraded forecast performance in the extratropics (Rodwell et al., 2018; Grams et al., 2018; Pickl et al., 2023; Berman and Torn, 2019). Therefore, the combination of processes (and the associated model uncertainty) that typically occur within WCBs make them a region where model uncertainties rapidly grow and amplify forecast errors (van Lier-Walqui et al., 2012; Morrison et al., 2020; Posselt and Vukicevic, 2010). This issue also increases the difficulty of quantifying the past, current and future radiative impact of WCBs (Joos, 2019).

Another critical factor, which has recently attracted attention, is the role of embedded convection within WCBs (Oertel et al., 2019, 2020; Schwenk and Miltenberger, 2024). Oertel et al. (2020) showed that convective regions within WCBs generate stronger surface precipitation, more intense diabatic heating, and have distinct impacts on the potential vorticity structure of the outflow. Schwenk and Miltenberger (2024) additionally showed that convective parcels transport significantly more ice

85

into the UTLS and undergo distinct microphysical processes compared to slower-ascending parcels. Furthermore, Hieronymus et al. (2025) and Schwenk et al. (2025) showed that regions of embedded convection have different parameter sensitivities compared to more slantwise ascending regions. The insight from these previous high-resolution, convection-resolving studies raises an important question: Can low-resolution, convection-parameterizing simulations (which are typically used for global forecasting and climate predictions) adequately represent WCBs, particularly regarding their moisture transport into the UTLS?

Several studies have shown that increasing model resolution from convection-parameterizing to convection-permitting scales can improve NWP model performance with respect to simulated top-of-atmosphere cloud-radiative effects (Senf et al., 2020) as well as precipitation patterns and the diurnal cycle (Vergara-Temprado et al., 2020). Choudhary and Voigt (2022) found that for simulations of WCBs, finer grid spacing resulted in more WCB air parcels ascending faster and reaching higher altitudes, alongside increased diabatic heating at all pressure levels, primarily from cloud-phase changes. This suggests that microphysical processes during WCB ascent are strongly resolution-dependent, likely tied to the role of convective air parcels (Schwenk and Miltenberger, 2024).

In this two-part study, we investigate the role of grid resolution in the modeled transport of water vapor and hydrometeors into the UTLS by WCBs. In Part-I (Schwenk and Miltenberger, 2025) we used Lagrangian online trajectories to examine how model grid spacing affects the microphysical processes and the transport of moisture and hydrometeors into the upper troposphere and lower stratosphere (UTLS) by a warm conveyor belt (WCB). Specifically, we analyzed two simulations of a WCB event over the North Atlantic on 23 September 2017, conducted with the ICOsahedral Nonhydrostatic (ICON) model: one convection-parameterizing (~13 km horizontal grid spacing) and one convection-permitting (~3.5 km grid spacing). We hypothesized, and our analysis strongly suggests, that the higher prevalence of larger vertical velocities in the high-resolution simulation is the driving force behind many of the differences between the two simulations.

Our results from Schwenk and Miltenberger (2025) indicate that the WCB outflow in the high-resolution simulation is drier. We attributed this to shorter relaxation timescales for supersaturation over ice driven by higher ice number concentrations. We also found differences in the microphysical processes experienced during ascent and in the fraction of frozen precipitation, which we plausibly linked to the stronger vertical motions in the high-resolution run. We provided a detailed insight into the microphysical processes and moisture conversion during ascent, because Schwenk and Miltenberger (2024) determined that these processes are sensitive to vertical velocities.

We also noted some important limitations of the Lagrangian analysis: the trajectory dataset is not domain-filling, meaning parts of the WCB may be underrepresented, and in the low-resolution simulation, the vertical motion of trajectories is not influenced by the convection parameterization scheme. In particular, the latter means we are only able to see the impact of convective moisture transport if a WCB trajectory passes over a grid-point where the convection scheme is triggered but not its impact on UTLS parcels not undergoing WCB ascent. Therefore, the aim of this second paper is expand our the analysis from

Schwenk and Miltenberger (2025) with an Eulerian analysis of the simulation data focusing less on the microphysical pathways in the WCB ascent but more strongly on the radiatively important UTLS moisture and hydrometeor content in the WCB outflow region. The second aim is to address the radiative implications: the drier WCB outflow in the high-resolution simulation could lead to more outgoing longwave radiation (OLR) at the top of the atmosphere (TOA), while the higher ice number concentrations could have the opposite effect. Our central research question for this paper is therefore: How does increasing model resolution from convection-parameterizing to convection-permitting affect the transport of moisture and hydrometeors into the UTLS, and how does it influence OLR at TOA?

The paper is structured as follows: First, we describe the methodology of the paper (section 2), which differs from Schwenk and Miltenberger (2025) in that we perform an Eulerian analysis as opposed to a Lagrangian analysis. We then briefly verify the results of Schwenk and Miltenberger (2025) from the Eulerian analysis (where possible; sections 3 and 4) and then analyze the differences in OLR at TOA and the causes of these differences (section 5).

### 2 Methods

This study is the second part of a two-part study. In Schwenk and Miltenberger (2025) and in this paper, we examine two simulations of one WCB case study, run at convection-permitting and convection-parameterizing resolutions. Furthermore, the case study, model setup (and simulation output) for the convection-permitting simulation are the same as in Schwenk and Miltenberger (2024), to which readers are directed to for details on the WCB case study and online trajectory analysis. Here, we summarize the key features of the case and setup in the following section. The subsequent sections describe the procedure of the Eulerian analysis, in particular our approach to identify WCB outflow airmasses from a Eulerian perspective.

# 2.1 WCB Case and ICON Model Setup

The WCB investigated in this study occurred over the North Atlantic on 23 September 2017, and we use the Icosahedral Nonhydrostatic (ICON) model (v2.6.2; Zängl et al. (2014)) for our simulations (see also Schwenk and Miltenberger, 2024). Both simulations were initialized with the operational ICON global analysis at 00:00 UTC on 20 September 2017, when the extratropical cyclone began to form off the coast of Canada, and ran for 96 hours until 00:00 UTC on 24 September 2017, by which time the WCB began to dissipate over Northern Europe.

Two simulations of this case at different resolutions have been conducted. The first simulation uses a R03B07 global grid which results in an approximate effective grid spacing of 13 km. This simulation is called the "global" simulation. The second simulation includes two higher-resolution domains that are nested into the global domain, and is therefore called the "nested" simulation. The nested regions were chosen such that they cover the main region of WCB ascent. The two nested domains use a R03B08 grid ( $\sim$ 6.5 km grid spacing) and a R03B09 grid ( $\sim$ 3.3 km grid spacing). Neighboring grids interact through two-way coupling (Zängl et al., 2022). The resolution of the highest-resolution domain was chosen to permit (deep and embedded)

130

convection explicitly, while shallow convection remains parameterized. In the global domain deep convection is parameterized using the Tiedtke–Bechtold scheme (Tiedtke, 1989; Bechtold et al., 2008). Sub-grid processes such as turbulence, orographic drag, and radiation, are parameterized using the standard ICON schemes (in all domains). For the parameterization of clouds microphysical processes, we use the two-moment scheme by Seifert and Beheng (2005). This scheme represents six hydrometeor species (cloud droplets, rain, ice, snow, graupel, and hail). In contrast to the nested simulation, the lower-resolution ("global") simulation only has one domain on the same R03B07 grid with an approximate grid spacing of 13 km. In this simulation, deep convection is parameterized by the Tiedtke–Bechtold scheme everywhere.

Although this study focuses on Eulerian simulation output, we use trajectory data as the basis to produce our Eulerian WCB masks (see Section 2.2). The trajectory data was produced by the online-trajectory module, Miltenberger et al. (2020); Oertel et al. (2023). In both simulations, trajectories were initiated at 3-hour intervals throughout the 96-hour simulation and output recorded every 30 minutes. The starting regions was selected to represent the WCB air stream as well as possible (Fig. 1) and starting positions were selected randomly from grid points within this region at six vertical levels spanning from approximately 1000 to 800 hPa. Selection of actual WCB trajectories from the full trajectory data-set was performed as described in Schwenk and Miltenberger (2025) and in Schwenk and Miltenberger (2024): Most importantly a WCB parcel has to ascend 600 hPa in no more than 48 h and be within a visually defined region at a specific time.

**Figure 1.** Simulation setup for the global simulation in a) and the nested simulation in b). Online trajectory starting area from Schwenk and Miltenberger (2025) is marked by the dashed lines; the nested domain 2 (red) and 3 (black) boundaries are indicated in b) by the solid lines. Data shows the ICON output for the total cloud cover and sea-level pressure on 23.09.2017 00:00 UTC.

## 2.2 Eulerian WCB mask

40 A direct, point-by-point comparison of the Eulerian output from the two simulations is neither feasible nor meaningful, since the simulated fields inevitably shift in time and space relative to each other. Therefore, we created an algorithm that can isolate






the WCB region in the Eulerian data for each simulation, using the WCB trajectory data as a basis. The goal is to produce masks that encompass as much of the WCB as possible and exclude non-WCB airmasses, and to then compare the simulations using a range of statistical measures, such as pressure-bin distributions and their mean and median. We only produce masks for the area encompassed by domain 3 (Fig. 1 b), because this is where the differences between simulations arise most strongly due to the differences in grid-spacing.

Our algorithm is split into two parts: First, we use the WCB trajectories to produce an initial two-dimensional (time-dependent) mask at each model output pressure level according to the following steps (schematic shown in Fig. 2):

- 1. At every pressure level, WCB trajectories that are located at pressures within  $\pm 25$  hPa of that level are selected (Fig. 2a).
  - 2. The number of identified trajectories at a given pressure level is represented on a Eulerian longitude-latitude grid (grid spacing 0.5 ° for both simulations; illustrated in Fig. 2 b). We restrict this to domain 3 for both simulations.
  - 3. The gridded product is normalized and smoothed using a Gaussian filter with  $\sigma = 1$  (Fig. 2c).
- 4. The smoothed gridded product is converted to a binary mask, where only grid points with a normalized trajectory count larger than 0.005 are retained as WCB grid points (Fig. 2 d).
  - 5. Finally, the WCB mask is re-gridded back to the original ICON grids for selecting the WCB related grid points in the native output of all simulations.

The resulting mask is very 'patchy' and leaves out large regions of the WCB where there are few or no trajectory points (Fig. 2 d). This reflects the limitation of the Lagrangian analysis presented in (Schwenk and Miltenberger, 2025) related to the non-domain filling properties of the Lagrangian data. Therefore, we add a crucial second step in creating the final Eulerian WCB outflow mask:

- 1. Small, isolated regions of the mask that are more than 350 (100) grid points away from the largest centroid of the mask for the nested (global) simulation are removed (see for example the bottom left region in Fig. 3 a). The aim is to remove as many outliers, i.e. regions not related to the main WCB outflow, as possible for the next step (even if they are again included in the mask at a later step).
- 2. The mask is repeatedly dilated (ca. 60 times in the global simulation and 300 times in the nested simulation; optimal number of iterations determined through trial and error). After each iteration, all grid points for which the specific humidity  $(q_v)$  is smaller than a custom threshold value for that pressure level are discarded. This allows the mask to slowly 'crawl' forward and fill in the WCB area without reaching into regions outside of the WCB, where  $q_v$  is much smaller (Fig. 3 b).
- 3. At the halfway point of iterations, the mask is smoothed using a Gaussian filter. This enables the mask to 'jump' across filaments of low  $q_v$  within the WCB area instead of having to go around them and removes small isolated regions of the mask that are surrounded by low  $q_v$  and have therefore not dilated.

**Figure 2.** Schematic for the first step of creating the Eulerian WCB masks using the WCB trajectory data: a) at each time step and for each model level, select trajectories close to model level, b) count the of number of trajectories per longitude-latitude bin (bin width of  $0.5^{\circ}$ ), c) use Gaussian filter to smooth the data, which is then normalized, and d) create binary mask using a threshold of normalized trajectory count and re-grid to original grid size. This figure serves as an illustration, therefore the grid boxes etc. are not to scale.

Figure 3.  $q_v$  at 400 hPa for the global simulation (top row) and nested simulation (bottom row) at 02:00 UTC on the 22. September, 2017. The area within the WCB mask is shown before (a, d) and after (b, e) the dilation. The the total  $q_v$  field in (c, f). Trajectory points within  $\pm 25$  hPa of this pressure level are plotted in cyan.

There is a marked improvement in coverage of the WCB area by the dilated mask (Fig. 3 b and e) compared to the initial mask (Fig. 3 a and d). The masks that we produced for both simulations during TF2 and TF3 are visually very similar in terms of area (when collapsed) and volume (Fig. 3 b and e; Fig. A2 b and e; otherwise not shown). The Area and volume of the masks are also similar in magnitude, but consistently larger in the nested simulation by about 5%, yet no more than 7% (not shown). This means that we should not expect substantial biases to arise from the WCB masks having different geographical coverage. Hence, the outlined algorithm can produce WCB masks that cover the entire WCB influenced area (and eliminate outliers) starting from non-domain filling trajectory data (and include outliers). In theory, one trajectory per pressure level could suffice to fill the WCB area at that pressure level through dilation. Therefore, the advantage of the masking procedure is that relatively sparse trajectory data can be used to produce the Eulerian WCB masks. Another advantage is that the resulting mask 'wraps around' the WCB in three dimensions, meaning it can exclude grid points that are below or above the WCB but not directly






part of it (Fig. A1 b). To obtain all longitude-latitude points that across which the WCB airstream passes at any altitude, the mask can be "collapsed" onto two dimensions (Fig. A1 a). In this way, two-dimensional variables such as total cloud cover or column-integrated rain mass can be retrieved.

Despite the overall benefits of the outlined algorithm to identify WCB influenced parts of the atmosphere, there are also some limitations. Although this method goes beyond trajectory data, it still relies on it. If there are too few trajectories at a given pressure level, regions of the WCB could still be excluded, even with dilation. The dilation procedure can also cause the mask to extend into unrelated weather systems that are geographically close to the WCB and also have a high  $q_v$ . This was particularly problematic for pressure levels close to the PBL, where  $q_v$  is generally high and the WCB 'touched' other mesoscale systems. The associated 'muddy'  $q_v$  field makes it difficult to prevent the masking procedure from dilating too far. For this reason, we only ran the second part of the masking algorithm for pressure levels below 600 hPa. For higher pressures, we only use the WCB mask after the first step (i.e. only based on Lagrangian data). However, as we are mostly interested in WCB outflow properties in our analysis, the impact of this on our analysis is expected to be small.

The outlined masking algorithm further relies on several human choices that make its automatic translation to other case-studies not straightforward. For instance, the precise implementation for this study included numerous custom thresholds, primarily determined 'by eye' (see the published code for specifics). Modifications were made to the  $q_v$  field (to prevent the mask from dilating into a certain region), the number of dilation iterations, and the final mask itself for certain pressure levels and time steps. This was an intensive process involving a lot of manual double-checking and is feasible for the case study presented here. However, for implementation in a climatological study, the algorithm would need to be adapted to ensure that the mask never dilates into unrelated weather systems because manually double-checking every time step would be too time-consuming. Nevertheless, the general approach we have conceptualized here (using trajectories and dilation according to level-specific  $q_v$  thresholds) could form the basis of an effective algorithm for this purpose. The benefits of this method compared to already established WCB detection algorithms such as from Quinting and Grams (2022), is that this process is inherently grid-scale agnostic and highly adaptable to be as inclusive or exclusive as desired (by tuning the respective thresholds). It also does not rely on pre-training a machine learning algorithm on a specific combination of variables, but instead only requires  $q_v$ .

# 210 2.3 Definition of time frames for analysis based on WCB ascent stages

In Schwenk and Miltenberger (2025) we defined three main time frames (TF) during which the WCB trajectories experience different parts of their ascent. During TF2, which is defined as beginning on 00:00 UTC 2017-09-22 and lasting to 06:00 UTC 2017-09-23, most trajectories begin and end their WCB ascent. Most of this ascent also occurs within the boundaries of domain 3 for both simulation. We will therefore focus on TF2 to investigate the WCB during the main ascent-phase. In TF3, defined as beginning on 00:00 UTC 2017-09-23 and ending at 20:00 UTC 2017-09-23, most trajectories remain above 500 hPa and are part of the WCB outflow. However, in TF3 many trajectories leave the boundaries of domain 3 in both simulations. In spite of

this, we restrict ourselves to domain 3 for the analysis of the outflow in TF3, because this is where the grid-spacings between simulations are the most different.

## 3 Differences in WCB properties during ascent-dominated stage (TF2)

## 3.1 Vertical velocity



WCB trajectories in the nested simulation experience more intense vertical motion overall, during both ascending and descending segments of the overall WCB ascent (Schwenk and Miltenberger, 2025). This suggests that the nested simulation has more chaotic and intense vertical motions. The Eulerian results clearly confirm this: For example, at 00:00 UTC 2017-09-23 and at 600 hPa, the masked vertical velocity (w) field in the global simulation shows large regions of coherently ascending vertical motion (Fig. 4a), whereas in the nested simulation the same field is more patchy with small regions of larger positive and negative vertical velocities (Fig. 4b). The distribution of w for this time and pressure is centered around  $0 \, \text{m s}^{-1}$  for both simulations, but for the nested simulation it shows longer tails to higher and lower values (Fig. 4c). Interestingly, the difference in the tail to negative w is greater between simulations than towards positive w, suggesting that the global simulation may not accurately represent descending motion in WCBs. Overall, this behavior is consistent across TF2 and across pressure bins (not shown).

Figure 4. Masked vertical velocities w at 00:00 UTC 2017-09-23 and 600 hPa for the global (a) and nested simulation (b). A normalized histogram of w at this time and pressure (c) is shown for the global simulation (red) and the nested simulation (black).





#### 3.2 UTLS moisture and condensate content

Our Eulerian investigation of the variables characterizing the ice hydrometeor population (i.e.  $N_i$ ,  $q_i$  and  $r_i$ ) reveals that changes to these variables when changing the simulation scale are broadly similar to those found in the Lagrangian investigation by Schwenk and Miltenberger (2025): Both  $N_i$  (Fig. 8) and  $q_i$  (Fig. A4) are larger in the nested simulation for pressures of 250 hPa or higher.  $r_i$  is much smaller in the nested simulation than in the global simulation and there is a sharp peak at small  $r_i$  for pressures of 400 hPa or lower, which is absent from the global simulation (Fig. 9). Schwenk and Miltenberger (2024) determined that convectively ascending air parcels contain ice crystals with much smaller  $r_i$  values than air parcels that ascend more slowly. Therefore, the peak at small  $r_i$  in the nested simulation can be very likely attributed to convectively ascending air, and its absence in the global simulation suggests that the convection parameterisation produces much larger  $r_i$  for detrained ice. This is unsurprising, because in the convection parameterization scheme, updrafts do not produce new ice-particles (which would increase  $N_i$  and introduce ice particles with small  $r_i$ ), but instead grow existing ice-particles (increasing  $r_i$ ).

Considering the vapor content, we determined in Schwenk and Miltenberger (2025) that at WCB outflow pressure levels, WCB trajectories have both a larger relative humidity over ice  $(RH_i)$  as well as higher specific humidity  $(q_v)$  in the global simulation compared to the nested simulation. In that paper, the detailed Lagrangian analysis suggests that this is most likely due to higher ice number concentrations  $(N_i)$  and smaller ice radii  $(r_i)$  in the nested simulation, which lead to smaller (faster) relaxation timescales for supersaturation over ice  $(\tau_{\text{sat,ice}})$ . We additionally speculated that larger  $q_v$  values in the global simulation could be due to the parameterized convective moisture and hydrometeor transport in that simulation. However, due to the non-domain filling nature and missing explicit representation of parametrized convective motion in the Lagrangian analysis, the importance of differences in explicit and parameterized water transport are difficult to assess from the Lagrangian perspective. Therefore, in this section we provide an overview of the moisture conditions in the UTLS during TF2 over the Eulerian WCB outflow and compare them with the results and hypotheses proposed in Schwenk and Miltenberger (2025) based on Lagrangian data.

Overall, the Eulerian analysis confirms the findings from Schwenk and Miltenberger (2025) with respect to the distribution of  $RH_i$  and  $q_v$  values (Fig. 5 and 6). From 200 to 250 hPa, the mean and median for  $RH_i$  are larger in the global simulation than in the nested simulation (Fig. 5 a and b). Additionally in the global simulation, about 30 % of grid points within the WCB mask have  $RH_i$  greater than 100 % at pressure levels between 200 to 250 hPa. This is consistently higher than in the nested simulation, for which this figure is around 25 %. For  $q_v$  the mean and median are larger in the global simulation at 200 and 250 hPa (Fig. 6 a and b), while they are almost identical both simulations at 400 and 450 hPa (6 c and d). At 250 hPa (Fig. 6 b), there is a pronounced third peak for  $q_v$  at approximately  $0.12 \,\mathrm{g\,kg^{-1}}$  in the global simulation that is absent in the nested simulation. These results indicate that in the global simulation, the UTLS outflow of the WCB is more moist on average, and also has a greater fraction of grid points that experience supersaturated conditions with respect to ice.

We note that we must be careful when deriving conclusions from the distributions of values, since grid-cells in the global simulation represent mean values for a larger area than in the nested simulation, meaning that there will be more extreme values in distributions of the nested simulation. However, this effect is not expected to change mean values, which we make sure to consider.

The geospatial distributions of  $RH_i$ , and especially of regions of supersaturation, are qualitatively similar in both simulations (Fig. 7). This indicates that the grid-resolution does not necessarily lead to large horizontal shifts or to completely different spatial moisture distributions overall. Instead, similarly to the distribution of w (Fig. 4 a and b), the key difference is the 'patchiness' of the  $RH_i$ -field; the global simulation exhibits large contiguous areas of supersaturation (Fig. 7 d-f), whereas in the nested simulation these areas are broken up into smaller fragments by regions that experience descending motion (Fig. 7 a-c). This additionally indicates that one reason for the smaller  $RH_i$  and lower frequency of supersaturated grid-cells in the nested simulation might be the larger abundance of descending motion (Fig. 4 c). This is an interesting result for those who study supersaturated regions in the UTLS (and for instance their fractal characteristics). We also note that the masking algorithm includes many areas where  $RH_i 

Figure 5. Normalized  $RH_i$  distributions at different pressure levels for the global (red) and the nested (black) simulations in TF2, with mean (solid) and median (dashed) values indicated by vertical lines.


We examine the same hypothesis from Schwenk and Miltenberger (2025) that larger  $N_i$  in the nested simulation can cause the smaller  $q_v$  and  $RH_i$ . While larger values for  $N_i$  decrease  $\tau_{\rm sat,ice}$ , smaller  $r_i$  do the opposite (Schwenk and Miltenberger, 2025; Schwenk et al., 2025). For pressures of 250 hPa and higher, the nested simulation has larger  $N_i$  and smaller  $r_i$  than the global simulation. However,  $N_i$  still determines the overall response of  $\tau_{\rm sat,ice}$ , because its increase is stronger than the decrease of  $r_i$  in the nested simulation (note: logarithmic x-scale in Fig. 8 and linear x-scale in Fig. 9): in the nested simulation  $\tau_{\rm sat,ice}$  is either similar or slightly smaller (Fig. A3), which could explain the smaller  $q_v$  and  $RH_i$ . However, in the highest part of the WCB outflow, i.e. 200 hPa, differences in saturation adjustment timescales due to different ice particle populations are not able to explain the shifts in the  $RH_i$  and  $q_v$  distribution.  $N_i$  is larger in the global simulation rather than smaller (Fig. 8 a), and  $RH_i$  as well as  $q_v$  is still larger than in the nested simulation. Instead, as we also hypothesized in Schwenk and Miltenberger (2025), the differences in  $q_v$  and  $RH_i$  at 200 hPa arise due to the convection parameterisation.

Figure 6. Normalized  $q_v$  distributions at different pressure levels for the global (red) and the nested (black) simulations in TF2, with mean (solid) and median (dashed) values indicated by vertical lines.

In TF2 and for the global simulation, 11% of grid-points in the collapsed WCB mask are influenced by convection. We define a grid-point as "convective" when the convective precipitation over the preceding 15 min exceeds  $0.01 \,\mathrm{kg}\,\mathrm{m}^{-2}$ . This definition holds only for that point in time, i.e, we do not "track" the convective areas. At 200 and 250 hPa, the convective data points in the WCB mask have a very different  $q_v$  to that of the non-convectively influenced data points: The former on average show

much larger  $q_v$  values (Fig. 10 a and b). As these areas account for a substantial proportion of the data points, it is obvious that this shifts the  $q_v$  distribution at 200 hPa to larger values (Fig. 6 a) and introduces a second maximum at  $q_v = 0.12 \,\mathrm{g\,kg^{-1}}$  at 250 hPa (Fig. 10 b, Fig. 6 b). For pressures of 400 hPa and larger, the data results in bi-modal  $q_v$  distributions with one peak at small  $q_v$  and another at large  $q_v$  (Fig. 10 c and d).



Figure 7.  $RH_i$  fields at 12:00 UTC 2017-09-22 for the nested simulation (top row) and the global simulation (bottom row) at 200 hPa (a and d), 300 hPa (b and e) and 400 hPa (c and f).

These differences in the  $q_v$  distributions for convective and non-convective regions clearly show that the convection parameterization produces moister conditions in the WCB outflow than regions where there is no convection (in that time step; a background signal from convection is expected in the "non convective" areas). In the nested simulation, where convection is not represented by a convection parameterisation, similar patterns cannot be identified, i.e. either the shift at 200 hPa nor a bimodality of the  $q_v$  distribution at higher pressures is evident. Schwenk and Miltenberger (2024) also identified substantial convective motion in the high-resolution simulation of this WCB case. Therefore, the differences in the  $q_v$  distributions suggests that the convection parameterisation overestimates the amount of moisture transported into the UTLS.

Figure 8. Normalized  $N_i$  distributions at different pressure levels for the global (red) and the nested (black) simulations in TF2, with mean (solid) and median (dashed) values indicated by vertical lines.

# 3.3 Cloudiness




The total, low, medium and high cloud cover values reflect the enhanced patchiness of simulated fields in the nested simulation (Fig. 11). The nested simulation experiences a greater proportion of grid points with cloud cover of any type greater than 95% than the global simulation. However, they also experience a greater proportion of grid points with total cloud cover less than 5%. This is not entirely unexpected, because grid points in the global simulation represent a larger area and therefore are more often only partly cloud covered, leading to less extreme values. The overall effect is that on average, the total cloud cover is slightly smaller in the nested simulation (76%) than in the global simulation (76.5%). More clear sky patches in the nested simulation, as well as different high, medium and low cloud cover values, might lead to differences in the outgoing longwave radiation between the two simulations.

Not only the cloud cover, but also the overall cloud content, represented by the total column integrated hydrometeor content, are different between the two simulations. In Schwenk and Miltenberger (2025), we investigated the Lagrangian hydrometeor content at different pressures and concluded that overall, the nested simulation produces more hydrometeors, which should result in thicker clouds. In that paper, we found the largest differences for graupel and rain, which are much abundant (in mass)



Figure 9. Normalized  $r_i$  distributions at different pressure levels for the global (red) and the nested (black) simulations in TF2, with mean (solid) and median (dashed) values indicated by vertical lines.

in the nested simulation. We attributed this to the higher vertical velocities in the nested simulation that allow for the retention of larger hydrometeors. For the Eulerian analysis, we do not investigate the hydrometeor content at pressure bins larger than 500 hPa, because in this lower region of the atmosphere the WCB masks are not produced using dilation (see Sec. 2.2) and therefore will mostly represent the trajectory data. Instead, we investigate the total column integrated hydrometeor mass for longitude-latitude points that lie within the collapsed WCB mask (i.e, all points that have at least one WCB grid-point above them; Fig. A1 a). Overall, the total column integrated hydrometeor mass (Fig. 12 c) is larger in the nested simulation. The difference is largest for rain (Fig. 12 b) and graupel (Fig. 12 c), and smallest for snow (Fig. 12 e). This is in line with our findings from Schwenk and Miltenberger (2025) and therefore indicates that the WCB in the nested simulation does indeed produce thicker clouds with more graupel and rain.

In summary, for the time frame during which most of the WCB ascent occurs (TF2), we were able to confirm many of the results from Schwenk and Miltenberger (2025):  $q_v$  and  $RH_i$  are larger for  $p \le 250$  hPa in the global simulation due to smaller  $N_i$ ;  $r_i$  is larger in the global simulation. These findings largely support the hypotheses we formulated in the beginning of this paper to explain the differences observed between the simulations. In particular, we are able to clearly identify the influence of


Figure 10. Normalized  $q_v$  distributions at different pressure levels for grid-points that experience convection (orange) and that do not (blue) in the global simulation only, in TF2. The mean (solid) and median (dashed) values are indicated by vertical lines.

the convection parameterisation on  $q_v$  in the global simulation, which was not possible in Schwenk and Miltenberger (2025). Our findings suggest that, although the Lagrangian trajectories did not fill the entire WCB area, the results obtained from them are largely valid for the bulk WCB outflow air mass. This lends confidence to the results from Schwenk and Miltenberger (2025) that cannot be verified from the Eulerian data, such as the accumulated microphysical process rates during ascent, and the results for the condensation ratio and precipitation efficiency. In the following we discuss TF3, the time frame during which most of the WCB trajectories are entirely part of the WCB outflow.

## 4 Differences in WCB properties during outflow-dominated stage (TF3)

In Schwenk and Miltenberger (2025), we established that in TF3, the trajectories of the global simulation remained slightly more moist than in the nested simulation at pressures below 250 hPa. In our Eulerian results for the same time-frame and pressure levels, we observe similar behavior for  $RH_i$  and  $q_v$ , even though this is not necessarily reflected in the mean and median values. For  $RH_i$ , the distributions at 200 and 250 hPa in the nested simulation have slightly longer tails for larger  $RH_i$  than the global simulation (Fig. 13 a and b). However, the peaks of the distributions in the global simulation are shifted slightly towards



**Figure 11.** Bar chart showing the percentage of grid points during TF2 that have high, low, medium and total cloud cover larger than 95% (a) and smaller than 5% (b) for the global simulation (red) and the nested simulation (black).

larger  $RH_i$  values (Fig. 13 a and b). At 400 and 450 hPa,  $RH_i$  distributions are similar in both simulations and bi-modal, but the second peak at smaller  $RH_i$  (indicating cloud-free regions; see Section 3) is larger in the global simulation, which results in smaller mean and median  $RH_i$  (Fig. 13c and d). For  $q_v$ , the distributions at 200 and 250 hPa (Fig. 14 a and b) are very similar to those in TF2 (Fig. 6 a and b), with the difference that  $q_v$  is no longer much larger in the global simulation. However, at 250 hPa the mean  $q_v$  remains slightly higher in the global than the nested simulation and interestingly the secondary peak at  $q_v = 0.12\,\mathrm{g\,kg^{-1}}$ , which was associated with the influence of the convection parameterisation (sec. 3) remains. At pressures of 400 hPa and higher, the mean and median  $q_v$  are similar but the peaks of the multi-modal distributions are shifted relative to each other (Fig. 14 c and d).

Similarly to TF2, we can attribute our observations for TF3 to the influence of the convection parameterisation, since the distributions for  $q_v$  in areas where the convection parameterization scheme is active are very similar to TF2 (Fig. 10). However, during TF3 and in the global simulation, convective grid points in the collapsed mask account for only around 6% of the data points, compared to 11% for TF2. This difference explains why the differences in mean and median values are not as pronounced, and implies that the differences in UTLS moisture conditions between simulations are largest when convective activity is strongest and is diluted in the following evolution likely due to mixing with the larger WCB outflow airmass, which is not affected by convective activity.



**Figure 12.** Normalized distributions of total column integrated hydrometeor mass in TF2 for cloud droplets (a), rain drops (b), total hydrometeor mass (c), ice (d), snow (e) and graupel (f). The mean (solid) and median (dashed) values are indicated by vertical lines.

Our Eulerian results for the ice content in TF3 do not necessarily reflect our Lagrangian results from Schwenk and Miltenberger (2025). In that paper, we determined that in TF3 both  $q_i$  and  $N_i$  are larger in the global simulation for pressures larger than 400 hPa, and smaller below (except for  $q_i$  at 200 hPa, which is larger in the global simulation). However, in our Eulerian results we find that  $N_i$  (Fig. 15) and  $q_i$  (Fig. A5) are slightly smaller in the global simulation from 200 to 450 hPa. This is unexpected because the data presented here for the global simulation takes convective grid points into account, and these produce much higher  $N_i$  and  $q_i$  in the UTLS than non-convective grid points (Fig. A6 and A7, respectively for TF2). However, one possible reason for the discrepancy in UTLS ice content between the Eulerian and Lagrangian analyses is that many in-situ cirrus clouds are probably formed in the WCB outflow and not observed by the trajectory data, similarly to Lüttmer et al. (2025).

Since the Lagrangian data does not fill the entire WCB area, we assumed that the effects of the convection parameterisation would be less noticeable in the Lagrangian trajectory data than in the Eulerian data. Therefore, we expected the Eulerian results to more strongly represent these grid points and to have higher  $N_i$  and  $q_i$  than the Lagrangian trajectories (at least compared to the nested simulation). However, in our results we find the opposite to be true: Lagrangian trajectories represent convective areas more strongly than the Eulerian data. In the 30-hour window defined by TF2, 50% of trajectories in the global

simulation spend over 10 hours above grid points flagged as convective in our mask (not shown). However, convective grid points only make up 11% of grid points in the Eulerian WCB mask during TF2. Therefore, this suggests that online trajectories in the global simulation represent convective grid points more strongly than the Eulerian data does. This probably occurs because regions of high vertical velocities, which enable trajectories to ascend as part of the WCB, coincide with areas where the convection parameterization is triggered. This could explain why  $N_i$  and  $q_i$  are larger at certain pressures during TF3 in the global simulation and the Lagrangian data from Schwenk and Miltenberger (2025), but not so in the Eulerian data presented here.


380

Another reason why the results for ice content in TF3 from Schwenk and Miltenberger (2025) are different may be that, during TF3, many trajectories extend beyond the boundaries of domain 3, whereas our Eulerian analysis is confined to this area.

Figure 13. Normalized  $RH_i$  distributions at different pressure levels for the global (red) and the nested (black) simulations in TF3, with mean (solid) and median (dashed) values indicated by vertical lines.

Figure 14. Normalized  $q_v$  distributions at different pressure levels for the global (red) and the nested (black) simulations in TF3, with mean (solid) and median (dashed) values indicated by vertical lines.

## 5 Results - Radiation

400

In Schwenk and Miltenberger (2025), we hypothesized that the differences in the vertical distribution of UTLS hydrometeor and vapor content between the two simulations might also be reflected in differences in effective cloud top temperature (CTT) as well as outgoing longwave radiation (OLR) at the top of the atmosphere (TOA). To test this hypothesis, we calculated the CTT by taking the collapsed two-dimensional mask at each time step and grid point, and setting the temperature at the highest model level at which the total hydrometeor mass exceeds 10<sup>-7</sup> kg kg<sup>-1</sup> as the CTT. We find that the differences in CTT between simulations are almost negligible for both TF2 and TF3 (Fig. 16 a and b). The distributions are very similar and the peaks are located at approximately the same temperatures. However, for TF2 (Fig. 16 a), the mean CTT of -49.5 °C in the global simulation is slightly smaller than the mean of -48.8 °C in the nested simulation. This is due to a slightly more pronounced second peak at approximately -60 °C in the nested simulation (Fig. 16 a).

The slight difference in CTT between the two simulations during TF2 is not able to explain the larger difference we find

Figure 15.  $N_i$  distributions at different pressure levels for the global (red) and the nested (black) simulations in TF3, with mean (solid) and median (dashed) values indicated by vertical lines.

for OLR (Fig. 16 b and c), which is clearly shifted to more negative values in TF2 in the nested simulation compared to the global simulation (meaning more outgoing radiation). The mean OLR in TF2 is  $-226.6\,\mathrm{Wm^{-2}}$  in the nested simulation, which is  $4\,\mathrm{Wm^{-2}}$  smaller than the mean of  $-221.5\,\mathrm{Wm^{-2}}$  in the global simulation. For the median, this difference is  $7\,\mathrm{Wm^{-2}}$ . During TF3, the mean values in both simulations are approximately equal at  $-223.7\,\mathrm{Wm^{-2}}$ , while the median is  $4.5\,\mathrm{Wm^{-2}}$  larger in the global simulation (opposite to before). However, this reversed difference is because the distributions of OLR in TF3 are very different (larger central peak for global simulation; Fig. 16 d), whereas the distributions in TF2 are more similar (Fig. 16 c). Therefore, when looking closely at the distribution of OLR values during TF3, we find that the edges still indicate a shift to more negative values for the nested simulation (same behavior as in TF2).

We determined that the large differences found OLR in both magnitude (TF2) and distribution (TF3) cannot be caused by differences in CTT, since these have nearly identical distributions as well as mean and median values (Fig. 16 a and b). Instead, the different OLR values must be caused by either differences in the UTLS vapor content or in the cloud fields. Upper tropospheric  $q_v$  can trap longwave radiation and act as a greenhouse gas. In TF2 and at pressures of 200 and 250 hPa,  $q_v$  is larger in

the global simulation than in the nested simulation (Fig. 6 a and b). Additionally, the peak for CTT is situated at approximately  $60 \,^{\circ}$ C, which is also approximately the mean temperature at the 200 hPa level during TF2 and TF3. This coinciding temperature means that  $q_v$  at pressures of 200 hPa can interact with the OLR radiating from the cloud tops at similar pressures. In TF3,  $q_v$  at pressures from 200 to  $450 \, \text{hPa}$  are neither substantially smaller or larger on average, but the distributions are shaped differently, with  $q_v$  in the global simulation exhibiting more pronounced peaks towards larger values (Fig. 14). This behavior fits to the detected differences in the OLR distributions: In TF2 OLR is more negative in the nested simulation (less greenhouse effect), while in TF3 differences are mainly detected in the shape of the OLR distributions for the two simulations (Figs. 16 c and d).

**Figure 16.** Normalized histograms of cloud top temperature during TF2 (a) and TF3 (b), and the outgoing longwave radiation at the top of the atmosphere during TF2 (c) and TF3 (d) for the nested (black) and global (red) simulation.

To determine whether the hypothesis that differences in OLR are primarily caused by differences in  $q_v$  is plausible, we first compare OLR inside of the WCB mask for clear sky conditions (defined as total cloud cover 

shifted to more negative values in the nested simulation (Fig. 17). In TF2, the mean clear-sky values are  $-279.0 \,\mathrm{W/m^2}$  and  $-282.5 \,\mathrm{W/m^2}$  for the global and nested simulation, respectively (difference of  $3.5 \,\mathrm{W/m^2}$ ). For TF3, the mean values are  $-283.0 \,\mathrm{W/m^2}$  and  $-288.7 \,\mathrm{W/m^2}$  for the global and nested simulation, respectively (difference of  $5.7 \,\mathrm{W/m^2}$ ). For TF2, the difference in mean clear-sky OLR between the two simulations is almost the same as the difference for total OLR. For TF3, the distributions of clear-sky OLR are very similar between the two simulations, whereas for the total OLR they are very different, indicating that differences in the cloud fields between the simulations must be important.

**Figure 17.** Normalized histogram of outgoing longwave radiation at the top of atmosphere for clear sky conditions (total cloud cover < 5%), for TF2 (a) and TF3 (b), nested simulation in black and global in red.

The comparisons of OLR under clear-sky conditions are compelling, however, a WCB is notoriously cloudy, which is why we must also compare OLR under cloudy conditions. To this end, we split the OLR and CTT data into two groups according to  $q_v$  at  $200\,\mathrm{hPa}$ ; the first group has  $q_v(200\,\mathrm{hPa}) > 0.035\,\mathrm{g\,kg^{-1}}$ , and the second  $q_v(200\,\mathrm{hPa}) \in (0.01,0.025)\,\mathrm{g\,kg^{-1}}$ . These values are chosed according to the distributions of  $q_v$  (not shown) and the lower bound is chosen to exclude clear-sky conditions. In both the global and the nested simulations and as long as CTT is between approximately -50 °C and -20 °C, OLR is much more negative (more outgoing LW radiation) for the second group, when  $q_v(200\,\mathrm{hPa})$  is smaller (Fig. 16). When CTT is smaller than -50 °C or larger than -20 °C, there is no difference in OLR between the groups. For CTT below -20 °C this could indicate the following: when cloud tops are situated at around 200 hPa, OLR does not depend on  $q_v$  at that pressure level, but instead on CTT. However, when cloud tops are situated below this level at higher pressures (i.e temperatures),  $q_v$  can trap the emitted

radiation. This clearly shows that differences in upper tropospheric  $q_v$  between the simulations could cause the differences detected in OLR. That the effect described here is stronger in the global that in the nested simulation once more indicates that the differences in the cloud fields are very likely also important for differences in OLR.

Figure 18. Outgoing longwave radiation at the top of atmosphere as a function of cloud top temperature, but split according to  $q_v$  at 200 hPa for the global simulation (a) and the nested simulation (b). Values for which  $q_v$  is larger than  $0.03\,\mathrm{g\,kg^{-1}}$  are plotted in orange, values for which  $q_v \in (0.01, 0.02)\,\mathrm{g\,kg^{-1}}$  in blue. Mean (solid) and median (dashed) lines are plotted, with the  $5^{\mathrm{th}}$  to  $95^{\mathrm{th}}$  percentiles shaded in light, and the interquartile range in dark.

### 6 Conclusions and Discussion

In this paper we investigate the role of simulation grid-scale for WCB moisture transport into the UTLS from a Eulerian perspective. In both Schwenk and Miltenberger (2025) and here, we examine a case study of a WCB that dominated the weather over the northern Atlantic on 23 September 2017, using two ICON simulations: one run at a convection-parameterizing grid spacing of ~ 13km ('global' simulation) and one run at a convection-permitting grid spacing of ~ 3.5 km ('nested' simulation). In the Schwenk and Miltenberger (2025), we analyzed Lagrangian trajectory data to investigate differences in cloud microphysical processes and their influence on moisture transport within a WCB. In the present study, we extend this analysis to the Eulerian model fields to examine the bulk characteristics of the WCB outflow, in particular, to assess the impact of the convection parameterization, which cannot be adequately addressed using Lagrangian data alone.

To analyze the WCB outflow in the Eulerian framework, we developed a simple and efficient approach to derive a WCB mask from trajectory data, allowing the outflow region to be isolated consistently at each pressure level (Section 2.2). To our

knowledge, this approach has not been used before. While the current implementation relies on several case-specific thresholds, it provides a useful foundation for future methodological refinements and could serve as an alternative or complement to the machine-learning-based WCB identification by Quinting and Grams (2022).

In Schwenk and Miltenberger (2025), we mainly attributed the differences in hydrometeor and vapor content in the WCB outflow between simulations to the different ascent velocities. The trajectories in the nested simulation exhibit much more rapid periods of ascent and descent, leading us to hypothesize that the nested simulation exhibits much more dynamic and chaotic vertical flow overall. Here, we verify this assumption by showing that, although the vertical velocity distributions within the WCB mask in both simulations are centered around zero, the distributions in the nested simulation extend to larger negative and positive values (Fig. 4). Higher vertical velocities allow for the retention of more and larger hydrometeors, which was the reason why trajectories in the nested simulation showed much higher total hydrometeor content and specifically more graupel during the ascent. With the Eulerian analysis presented here we were able to generalize this for the entire WCB by examining the total column integrated hydrometeor content underneath the WCB mask (Fig. 12, which is larger overall in the nested simulation, and much larger in the case of graupel and rain.

Regarding UTLS moisture, Schwenk and Miltenberger (2025) found that the WCB outflow in higher regions of the atmosphere is more moist in the global simulation than in the nested simulation. We attributed this to the different  $N_i$  values, which were much higher in the nested simulation, allowing for a faster reduction of supersaturation by reducing the relaxation timescale of supersaturation over ice  $\tau_{\rm sat,ice}$ . In this study, we also find that in the global simulation, the WCB outflow is more moist at pressures of approximately 350 hPa and below. This was observed for both the specific humidity (Fig. 6) and the relative humidity over ice (Fig. 5). For pressures larger than 200 hPa, this can be attributed to higher  $N_i$  as in Schwenk and Miltenberger (2025) (Fig. 8). However, at 200 hPa this explanation is not valid, because  $N_i$  is not higher in the nested simulation, but the differences in vapor remain.

We therefore identified a second mechanism to explain the differences in humidity: the influence of the convection parameterization in the global simulation. In the global simulation, regions within the WCB mask that exist over areas of convective precipitation have much higher  $q_v$  values than all other points within the WCB mask (Fig. 10). If the representation of upper-level moisture and hydrometeor content in a convective atmospheric column by the convection parameterization were consistent with that in an explicitly resolved convective column, there would be no difference in upper-level moisture between the two simulations. However, Schwenk and Miltenberger (2024) found that convective ascending air in a convection permitting simulation produces very high  $N_i$  and thus small  $\tau_{\rm sat,ice}$ , leading to  $RH_i$  values that peak at about 100%. This results in smaller upper-level moisture content for convective ascending parcels, not more. Therefore, the difference in convectively ascending air between the convection-parameterizing global simulation and the convection-permitting nested simulation is another factor contributing to differences in upper-level moisture.






In the global simulation, the convection parameterization produces larger ice-crystals in the WCB outflow than one would expect given the results from Schwenk and Miltenberger (2024), which show that fast ascending air produces smaller ice crystals than slower ascending air. The distribution of  $r_i$ -values for the global simulation lacks the bi-modality that is prominent in the nested simulation, especially for pressures of 350 hPa and smaller (Fig. 9). Assuming that a simulation that allows for convection better represents reality than one using a convection parameterization, the results presented here for upper-tropospheric moisture and hydrometeor content suggest the latter fails to accurately simulate moisture transport into the UTLS.

After the primary ascent phase has passed, we find slightly different results concerning UTLS ice content than in Schwenk and Miltenberger (2025), which could be because the Eulerian data captures in-situ cirrus clouds that are not captured by the Lagrangian trajectory data. We also find that it is possible that the trajectory data from the global simulation over-represents areas of convection. This is because the Eulerian analysis, which should include most convective regions in the WCB that are not represented by the Lagrangian data, also includes many non-convective regions that are most likely not represented by the trajectory data.

The two simulations show similar cloud-top pressures and temperatures, but distinct spatial patterns of outgoing longwave radiation (OLR) in the WCB outflow region (Fig. 16). Although global climate models are typically tuned to achieve radiative balance under pre-industrial conditions, this tuning primarily affects absolute OLR values, not regional variations. The differences we find—more OLR in the nested simulation over the masked WCB region—therefore highlight physically meaningful regional contrasts. Since cloud-top temperatures remain comparable, the enhanced OLR must arise from reduced UTLS vapor in the nested simulation, consistent with our analysis of  $q_v$  at 200 hPa (Fig. 18) and the clear-sky OLR differences. This suggests that convection-parameterizing simulations may misrepresent regional radiative effects of WCBs, even if their global energy balance remains constrained by tuning.

In summary, in this paper we (i) developed a novel algorithm for creating WCB masks in Eulerian data based on Lagrangian trajectories; (ii) showed that simulations that parameterize convection may be substantially biased with respect to the WCB transport of moisture into the UTLS, and (iii) demonstrated that a purely Lagrangian analysis, in which detection of coherent airstreams is straightforward, produces similar results to an Eulerian analysis, which faces less sampling issues. This means that future studies examining only Lagrangian data in WCBs can be (carefully) considered robust in representing the entire WCB, as long as the data is not too sparse.

Our results show that the convection parameterization produces WCB outflow that is too moist, with too few ice crystals that are too large, when compared to the higher resolution simulation. This indicates that global climate models might overestimate the transport of moisture into the UTLS and incorrectly depict the properties of cirrus clouds in WCB outflows. Given the assumption that a future increase in sea surface temperatures will increase the occurrence of convection, this bias could increase further still. A larger study comparing WCB simulations at different resolutions could therefore be worthwhile. Such

a study could compare atmospheric columns in convective areas between simulations and aim to improve the representation of vertical transport of moisture and representation of cirrus clouds by the convection parameterization scheme.

Code and data availability. Code and data will be made available after publication and will appear under the TPChange community on zenodo: https://zenodo.org/communities/tpchange.

Author contributions. CS and AM designed the experiment and conducted the numerical simulations. CS wrote the post-processing code. CS and AM worked jointly on the interpretation of the results. CS drafted the final manuscript with contributions from AM.

Acknowledgements. This work was funded by the Deutsche Forschungsgemeinschaft (DFG, German Research Foundation) through TRR 301 (project no. 428312742; The Tropopause Region in a Changing Atmosphere) sub-project B08 coordinated by Annette Miltenberger. The authors gratefully acknowledge the computing time granted on the supercomputer MOGON 2 at Johannes Gutenberg University Mainz (https://hpc.uni-mainz.de/ last access: 9 December 2024), which is a member of the AHRP (Alliance for High Performance Computing Rhineland-Palatinate, https://www.ahrp.info/, last access: 9 December 2024) and the Gauß-Allianz e.V. We further thank Annika Oertel for useful discussion input and sharing her ICON setup used in Oertel et al. (2023).

# **Appendix A: Additional Figures**

**Figure A1.** Example depiction of how the Eulerian WCB masks (black is false, yellow is true) "wraps" around the WCB trajectory positions (cyan) as seen from above (a) and from the side (b) at the model run-time of 144 h (00:00 UTC on the 23. September, 2017) for the nested simulation within the boundaries of domain 3.

Figure A2. Heatmaps showing  $q_v$  at 400 hPa on 00:00 UTC 23 September 2017 in the global / nested simulation for domain 3 masked before dilation (a / d), masked after dilation (b / e), and unmasked (c / f). Cyan points indicate trajectory positions in the vicinity of the 400 hPa pressure level.

Figure A3. Normalized  $\tau_{\text{sat,ice}}$  distributions at different pressure levels for the global (red) and the nested (black) simulations in TF2, with mean (solid) and median (dashed) values indicated by vertical lines.

Figure A4. Normalized  $q_i$  distributions at different pressure levels for the global (red) and the nested (black) simulations in TF2, with mean (solid) and median (dashed) values indicated by vertical lines.

Figure A5. Normalized  $q_i$  distributions at different pressure levels for the global (red) and the nested (black) simulations in TF3, with mean (solid) and median (dashed) values indicated by vertical lines.

Figure A6. Normalized  $q_i$  distributions at different pressure levels for grid-points that experience convection (orange) and that do not (blue) in the global simulation only, in TF2. The mean (solid) and median (dashed) values are indicated by vertical lines.

Figure A7. Normalized  $N_i$  distributions at different pressure levels for grid-points that experience convection (orange) and that do not (blue) in the global simulation only, in TF2. The mean (solid) and median (dashed) values are indicated by vertical lines.

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
