# Peer review of "Effects of Model Grid Spacing for Warm Conveyor Belt (WCB) Moisture Transport into the Upper Troposphere and Lower Stratosphere (UTLS) — Part II: Eulerian Perspective"

_EGUsphere, 2025_

## Referee Comment (RC1)

**Effects of Model Grid Spacing for Warm Conveyor Belt (WCB) Moisture Transport into the Upper Troposphere and Lower Stratosphere (UTLS) – Part II: Eulerian Perspective, Cornelis Schwenk and Annette Miltenberger**

C O N T E N T S

**SUMMARY**

In this manuscript, Schwenk and Miltenberger present a North Atlantic warm conveyor belt (WCB) case study comparing moisture transport, hydrometeor properties, and radiative effects between convection-permitting and convection-parameterizing simulations. A key contribution is a novel algorithm that combines Lagrangian trajectory information with moisture fields to construct three-dimensional WCB masks, which are then used for an Eulerian analysis of WCB outflow regions.

The studies findings – higher moisture, fewer but larger hydrometeors, and reduced outgoing longwave radiation (OLR) in the WCB outflow of the convection-parameterizing simulation – are highly relevant for understanding WCB-related biases in upper-level dynamics and radiative budgets in climate models.

I particularly commend the authors on their innovative approach to create WCB masks and the generally clear and well structured presentation. The effort to diagnose and interpret the processes underlying the simulated differences is especially valuable, as it points toward potential pathways for improving convection parameterizations.

I look forward to seeing the study published after one major issue is addressed. This concerns the treatment of differing model resolutions. Using data at different resolutions both in the WCB detection algorithm and in the subsequent analysis raises the risk that some of the diagnosed differences reflect sampling effects rather than physical processes. As a result, it possible that parts of the results are influenced by sampling-related biases. Addressing this issue would substantially strengthen the line of argument in several parts of the results section.

I detail this resolution concern, along with additional general comments, below, followed by specific comments on individual sections and figures.

**1 | GENERAL COMMENTS**

As noted above, I have one major concern, which relates to the treatment of simulation output at differing horizontal resolutions. As I understand the setup, the authors analyse output at 3.5 km resolution for the nested and at 13 km resolution for the global. Consequently, one grid point in the global simulation corresponds to roughly nine grid points in the nested domain, with clear implications for the sampling and statistical representation of quantities that vary on small spatial scales.

The authors acknowledge this issue in the manuscript, but its implications are not addressed sufficiently in the analysis. In my view, this is problematic because resolution-related sampling differences may influence both the identification of WCB regions and the statistical properties diagnosed within them in an unintended manner. Below, I outline two specific ways in which this resolution mismatch could affect the results and interpretation, and which I believe should be addressed to strengthen the robustness of the study's conclusions.

1. **Differences in WCB masks partly due to sampling?**

   The WCB mask algorithm is elegant and provides a valuable way of evaluating a Lagrangian feature in an Eulerian framework. However, I am concerned that the use of original-grid data at different resolutions in the WCB detection algorithm may introduce sampling effects that influence the resulting WCB masks.

   In particular, I wonder why the authors do not use moisture fields ($q_v$) regridded to the resolution of the global simulation as a common basis for identifying WCB masks in both simulations, which would seem more consistent to me. The finer grid of the nested simulation inevitably exhibits greater small-scale variability in $q_v$. As a result, regions in which high- and low-$q_v$ grid points coexist at high resolution may yield different mask extents than the same fields after regridding, even if the large-scale structures were identical between simulations. Under the current approach, this alone could lead to systematic differences in WCB masks. This concern is particularly relevant for high-level WCB masks, where a more spatially dispersed WCB with diffuse edges may lead to the inclusion of grid points with relatively low $q_v$ in the high-resolution masks. While I

am not certain to what extent this effect is quantitatively important, Figure 3 is at least consistent with this interpretation, as a small gap in the high-resolution WCB mask corresponds to a much larger non-WCB area in the global simulation. The systematic differences found in WCB-mask size make me particularly interested in this effect: Maybe, part of the difference in size comes from the different grid spacing?

In addition, the differing resolutions necessitate the use of two slightly different versions of the mask-generation algorithm, with different numbers of iterations that are determined by trial and error for each resolution. Regridding the high-resolution data prior to mask generation would allow the same algorithmic settings to be applied to both simulations, thereby removing one subjective element from the procedure.

A final, more conceptual concern relates to the use of a threshold in a quantity that is itself analysed within the WCB regions. Since WCB impacts on humidity and hydrometeor properties are a central focus of the study, it is not obvious that defining WCB membership based on a fixed $q_v$ threshold is entirely unproblematic.

In summary, I am concerned that the use of original-grid output for WCB mask generation may introduce resolution-dependent sampling effects. If the authors can convincingly argue that these differences are physically meaningful or even desirable, this concern may be alleviated. Otherwise, I suggest testing the sensitivity of the WCB masks to the use of regridded data and, if necessary, repeating parts of the analysis based on this more consistent masking approach.

2. **Sampling effects in the evaluation of WCB-area properties** Beyond the identification of WCB regions, similar sampling issues may also affect the evaluation and interpretation of WCB-area properties. For much of the analysis, the authors rely on output at the original model resolutions, even when discussing not only mean values but also spatial structure and full distributions (see, for example, the discussion around line 265 ff.). For quantities with pronounced small-scale variability, differences between the simulations may therefore partly reflect differences in sampling density rather than genuine differences in physical behaviour.

In particular, differences in spatial structure or in the width and tails of distributions may arise simply because the higher-resolution simulation resolves more small-scale variability, rather than because the underlying processes are fundamentally different. This is relevant not only for the interpretation of distribution tails, but also for dynamically important variables for which scale interactions play a role. One example is vertical motion, which is discussed in connection with Figure 4, where finer resolution alone could lead to apparently stronger variability or more extreme values.

A possible way to address this issue would be to complement the existing analysis with diagnostics based on regridded data, in addition to the original-resolution fields for those cases where sampling effects are likely to influence the conclusions, as already acknowledged by the authors around line 265 ff. This would allow the authors to assess how the higher-resolution simulation behaves at spatial scales that are also represented in the coarser simulation, and to better separate resolution-related sampling effects from physically meaningful differences.

Such an approach could also strengthen the interpretation of results for hydrometeor number concentrations, Since the central argument concerns aggregated effects of convection-permitting versus convection-parameterizing behaviour – rather than purely stochastic or noisy variability associated with explicit convection.

Some more general, but less critical points:

1. This is more of a framing-issue: You consistently take the convection-permitting simulation as a kind of ground truth. While this is understandable, I would consider wording the comparison a bit more neutral due to the lack of an actual ground truth to which to compare the simulations.

2. Relevance of the findings for NWP: I find the results on UTLS humidity very interesting from an atmospheric dynamics point of view. Maintaining tropopause sharpness hinges on producing accurate humidity gradients and representation of radiation at the tropopause, such that the presented results are relevant not only for climate- but also weather modelling. I

encourage the authors to mention this in their abstract as well as discussion to point colleagues in the dynamics community toward the relevance of this work.

3. My last general question concerns the choice of analysed domain: The authors choose only nested-domain trajectories, specifically for TF3. I don't quite understand why the resolution is so critical here – if the aim is to study the impact of the convection parameterization on WCB-outflow. If the air parcels are mostly just traveling at high levels in this time frame, it doesn't seem too important that the grid size is the same at this stage? Shouldn't the effects of explicit convection (even if in a mixed manner) survive the air parcels traveling outside the high-res domain?

**2 | SPECIFIC COMMENTS**

**| Abstract**

Line 6: Here, *we* present ...

Line 17 ff: "variations in water vapor concentration drive one of the most important positive climate feedbacks (Li et al., 2024; Held and Soden, 2000; Dessler et al., 2013, e.g.,),"

Why not briefly mention which feedback mechanism this is?

Line 35 ff: There is a mixing of singular and plural formulations regarding WCBs here. Consider rewriting this for clarity?

Line 90: The sentence starting with "Therefore" has some grammar problems and is rather lengthy. Maybe split into two?

**| Methods**

Line 134: 'The starting regions *were* selected ..' or both singular

Line 139: 'visually defined region' does not really help me understand your criterion here. It would help if that would be a bit more specific (which I think would be good, even though you describe the method in a detailed manner in Part I.)

Line 184: There is a 'that' too much.

Line 214: this should be 'for both simulation*s*'.

**| Results**

Here, I thought whether it might be useful to generally clearly delineate between things that are in line/ in (apparent) contradiction with findings from the Lagrangian study, as well as completely new aspects that could only be found in the Eulerian analysis. You do this in parts of this section, but might be a bit more consistent in doing so.

**Line-by-line comments:**

Line 228: With Results being consistent across TF2 you mean spatially / temporally/ both?

Line 234: You talk about Figure 8 here, before Figure 5-7. Consider moving Figure 8 up, so the reader doesn't need to scroll that much?

Line 232: How about briefly defining $N_i, r_i, q_i$?

Line 260: 'while they are almost identical *in* both simulations ...' ?

Line 321: 'Which are much *more* abundant' ? Line 410: 'The large differences found *for* OLR' Line 415: I think there is a minus sign missing in front of the temperature. Line 434 f.: 'These values are *chosen*' line 424: 'That the effect described here is stronger in the global *than* in the nested'

**| **Conclusions and Discussion**

Line 449: Remove 'the' ?

Line 468: Paranthesis missing after (Fig. 12

Line 497 ff: I would appreciate if this comparison to the first part was more concrete. How about briefly stating the difference again, so people don't need to jump between papers to understand (especially the) discussion? Additionally I find the argument you make in line 500ff somewhat hard to follow. Maybe you try to rephrase it such that the connection between the first and second sentence of the argument becomes more clear?

Line 505 ff: I think you somewhat understate the relevance of your results for climate modelling. The models are tuned to be balanced for present-day climate, but as you mention later, if the overall role of convection changes in a warmer atmosphere, the mis-representation of radiative effects of convection might lead to biases in the energy budget of the planet. Maybe you could consider merging your last paragraph, where you discuss this, to this paragraph.

Lastly, I think that your findings are relevant not only for climate modelling, but also for NWP, where we know that the humidity-gradient at the tropopause is relevant for accurately forecasting jet intensity and RWB evolution (e.g. Gray et al., 2014; Saffin et al., 2017) and hence forecast fidelity.

**| **Figures**

Figure 1: Caption misses description of colored contours, likely SLP. The dashed black lines are kind of hard to see, especially given the also black coastlines. Maybe a different color (a light grey?) could distinguish it more clearly from other features?

Figure 3: Sentence 3 has a 'the' too much in the beginning. You could also consider noting 'global' and 'nested' simulation beside the respective rows, to make the figure even more self-explanatory.

**REFERENCES**

Gray, S. L., Dunning, C., Methven, J., Masato, G. and Chagnon, J. M. (2014) Systematic model forecast error in Rossby wave structure. *Geophysical Research Letters*, **41**, 2979–2987.

Saffin, L., Gray, S. L., Methven, J. and Williams, K. D. (2017) Processes maintaining tropopause sharpness in numerical models. *Journal of Geophysical Research: Atmospheres*, **122**, 9611–9627.